# Assessing trauma center accessibility in the Southeastern region of the U.S. to improve healthcare efficacy using an anti-covering approach

Heewon Chea[ID]*[¤], Hyun Kim[ID]

Department of Geography, University of Tennessee, Knoxville, Tennessee, United States of America

¤ Current address: Center for Transportation Research (CTR), University of Tennessee, Knoxville, Tennessee, United States of America

* hchea@vols.utk.edu

## Abstract

Accessibility to trauma centers is vital for the patients of severe motor vehicle crashes. Many vehicle crash fatalities failed to reach the proper emergency medical services since the accident location was far away from trauma centers. The spatial discordance between the service coverage area of trauma centers and actual locations of motor vehicle accidents delays the definitive medical care and results in death or disability. Many fatalities would have been prevented if the patients had a chance to get proper treatment in time at Southeastern region of the U.S. Also, the accessibility to trauma centers from the actual locations of motor vehicle accidents is different in the Southeastern region. This research aimed to facilitate the accessibility to trauma centers for severe motor vehicle crash patients in the Southeastern region. The analyses are conducted to assess current trauma center accessibility and suggest the optimal locations of future trauma centers using the Anti-covering location model for trauma centers (*TraCt* model). This study found that existing trauma centers failed to serve many demands, and the actual coverages of the current locations of trauma centers over potential demands are highly different in each Southeastern state. *TraCt* model is applied to each Southeastern state, and its solutions provide better coverage for demand locations. However, the *TraCt* model for each state tends to choose too many facilities, with excessively supplied facilities across the Southeastern region. The excessive service supply issue is addressed by applying the *TraCt* Model to a broader spatial extent. *TraCt* model applied to the entire Southeastern region and most of the demand, over 98% covered by the service coverage of optimal facility locations with 15 additional facilities. This research proves that the GIS and *TraCt* model applied to the broader spatial extent works well with increasing trauma medical service beneficiaries while providing a minimum number of additional facilities.

**Data Availability Statement:** Data processed for this study is shared by GitHub and open to the public without restriction. The link for accessing

the database is as below. https://github.com/
Neologic21/PGPH-D-22-01274.

**Funding:** This research was partially supported by
the National Science Foundation (BCS-1951344).
Any opinions, findings, conclusions, or
recommendations expressed in this article are
those of the authors and do not necessarily reflect
the views of the National Science Foundation.

**Competing interests:** The authors have declared
that no competing interests exist.

## Introduction

The spatial disparity in trauma center accessibility is an understudied problem in the Southeastern region of the United States. Many motor vehicle accidents occur outside the 60-minute travel time distance coverage area of trauma centers. Furthermore, the number and the spatial distribution of trauma centers differ in each state, which has affected the overall volume of demand covered by trauma centers across the 12 Southeastern states, as of 2019. Because having access to trauma centers is crucial in life-or-death situations [1–4], the severe inequality of this access should be alleviated across the region to improve healthcare equity. Trauma centers should be strategically placed to cover the maximum number of potential demands. This study aims to assess the spatial disparities of trauma center accessibility in the Southeastern states of the United States and provide definitive policy recommendations based on the results of analysis using geographic information system (GIS) and solving the location problem.

This research has three specific objectives to improve the location of future trauma centers. The first is to measure the current levels of potential demand coverage from existing trauma centers in Southeastern states. The healthcare inequality issue can be understood by reviewing the different coverage of fatalities and the number of potential demand locations among the 12 Southeastern states. The second is to try to find the optimal location for trauma centers based on the actual locations of motor vehicle crashes. A quantitative method, the *Phi* correlation coefficient, is applied to determine the spatial disparity in accessing trauma centers from potential demand locations by comparing the different numbers and spatial distributions of trauma centers. The third is to find the optimal location for trauma centers by changing the spatial extent, with the results differing according to the scales of the study area. The accumulation of solutions for each state and the general solution for the whole Southeastern region are then compared to find a better spatial scale of the study area.

This research consists of three analyses: The first part compares the number and percentage of potential demands of each state covered by the current trauma system in the Southeastern region. The second analysis uses the anti-covering location problem for trauma centers (*TraCt* model) presented by Chea *et al.* [5] to locate potential trauma centers with better coverage for vehicle crashes with fatalities. The *TraCt* model is applied to each state to find the optimal location of trauma centers. The accessibility to trauma centers from demand locations are then compared in two different scenarios: the coverage of the chosen facilities from the *TraCt* model, and the coverage of current trauma center locations. In the final part of this study, the *TraCt* model is applied to the entire Southeastern region to compare demand coverage and efficiency in the selected facilities to the aggregated total solutions for each state of the 12 Southeastern regions.

## Background

### Spatial disparity of accessing trauma centers in Southeastern states of U.S.

Prior research has been focused on spatial disparities in accessing trauma centers in rural areas [2,6,7]. Some of these research has focused on the Southeastern region of the U.S. and found that it has poor trauma accessibility compared to the other regions in the country. However, despite these insight, prior research has overlooked the fact that even within the Southeastern region there are different levels of accessibility to trauma centers. According to the region classification of the Bureau of Economic Analysis (BEA), the 12 states comprising the Southeast region are Alabama (AL), Arkansas (AR), Florida (FL), Georgia (GA), Kentucky (KY), Louisiana (LA), Mississippi (MS), North Carolina (NC), South Carolina (SC), Tennessee (TN), Virginia (VA), and West Virginia (WV) [8].

Vehicle crashes are the second highest reason for patients being transferred to the nearest trauma centers [9]. Since serious vehicle accidents are likely to cause severe injury or death to patients, it is crucial to get emergency medical care as fast as possible. Depending on the severity of the damage, emergency surgery may be required, and the distance from the accident site to the trauma center is often important in determining the life or death of a patient.

Researchers have shown that in cases where severe vehicle accidents have resulted in fatalities, patients failed to reach the nearest trauma centers or emergency medical care facilities in time [10,11]. However, only a little research has been done concerning the accessibility of trauma centers from the vehicle accident location to the nearest trauma center.

## Spatial discordance between the distribution of demands and administrative area

The spatial distribution of vehicle accidents does not always occer adjacent to the clusters of the population distribution. And many vehicle accidents happen near or crossing the administrative boundary between the different states. Vehicle crashes happen alongside the road network, and there are specific locations where more vehicle accidents occur regardless of the state's administrative boundary [12]. However, the trauma centers are usually located in or adjacent to urban areas with large numbers of residents or floating populations.

The problem is that many severe accidents do not occur near urban areas and so have lower chance of accessing emergency medical facilities. Many severe car crashes take place outside of cities, near state borders or in rural areas. As a result, many patients from accidents cannot get to the emergency medical care facility on time [11]. This makes it imperative to consider accessibility to trauma centers from the actual locations of vehicle accidents to find the optimal placement for trauma centers.

## Data

### Candidate facilities: Trauma centers and general hospitals

The candidate facilities in this study are the trauma centers and general hospitals in the Southeastern region that operated 24 hours a day of every week in 2019. As a practical, evidence-based approach, this model works with these candidate location features to find the best locations for trauma centers. The locations of trauma centers in the twelve Southeastern states are available from each state's Department of Health website. The data source for the actual location of trauma centers in 2019 was released the following year. However, as some states do not provide the list of trauma centers every year, the most recent information for each state has been used. The locations of general hospitals in Southeastern regions can be accessed via the location information or mailing address of the Centers for Medicare and Medicaid Service in 2019 [13].

The operational definition of travel time distance constraints (TDC) is used to define the coverage from each trauma center and candidate location based on the time geography concept of potential path area [14]. As the standard time frame in which patients should be transferred to the nearest emergency medical care facility, a 60-minute travel time constraint is applied to the TDC of this research to show the coverage of each facility [15,16].

Fig 1 shows the spatial distribution of motor vehicle crashes in the study area, the locations of trauma centers of 2019, and its 60-minute TDC in the Southeastern region as of 2019. The number of current trauma centers is 226, and there were 63,946 fatalities from 2015 to 2019 in the overall Southeastern region. The 56,846 fatalities (88.9%) are within the 60-minute TDCs from trauma centers. The uneven spatial distribution of trauma centers by state is noticed

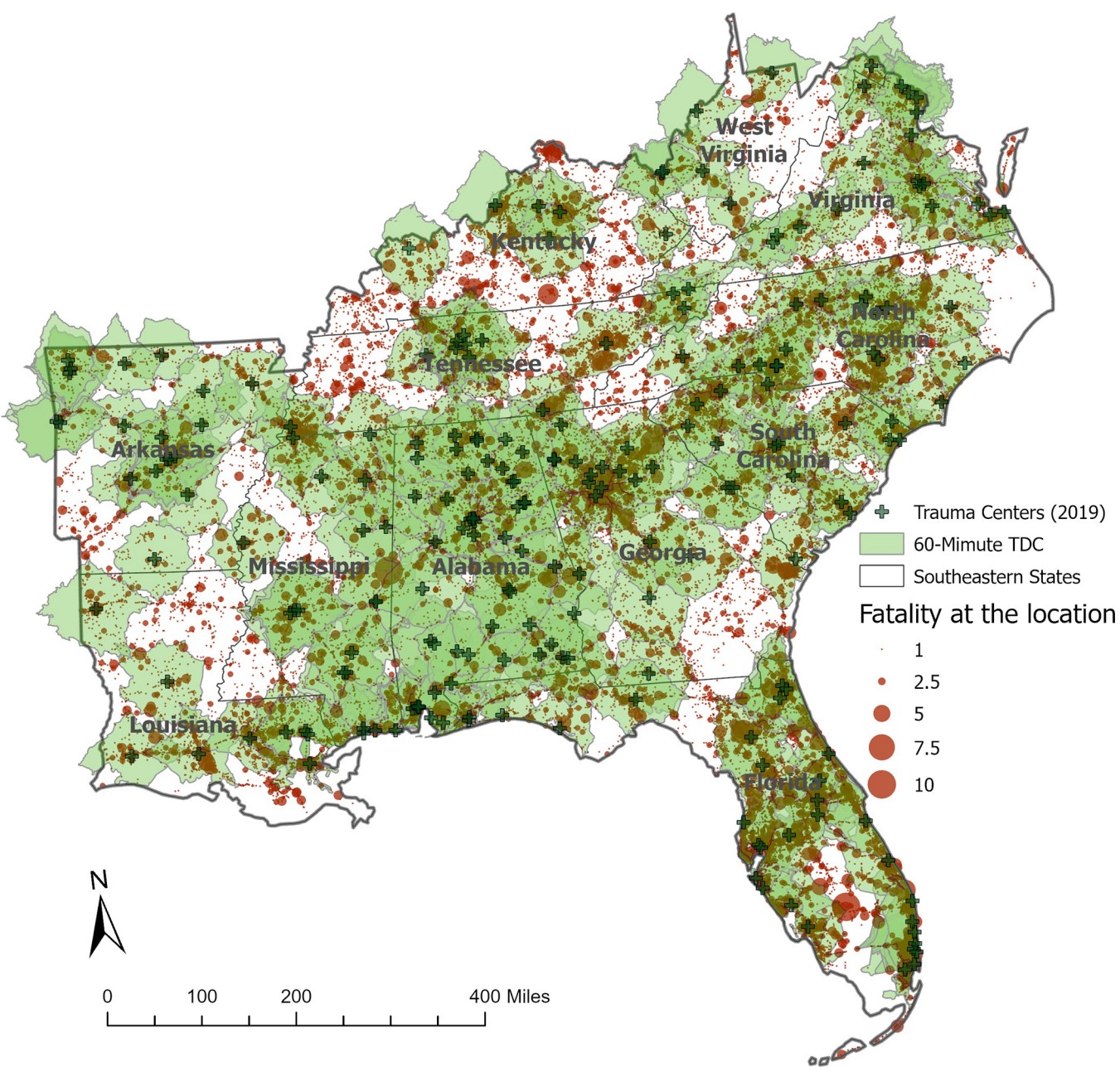

**Fig 1. Spatial distribution of trauma centers' 60-minute TDCs of 2019 and fatality of motor vehicle accidents in the Southeastern region.**

based on visual inspection. The locations of trauma centers and motor vehicle accident fatalities in study areas failed to correspond with an hour TDC completely, indicating the disparity of trauma medical services in the Southeastern region.

## Data process on producing potential demand locations

The number of fatalities resulting from motor vehicle crashes is considered the potential demand should be transferred to the nearest trauma center in this study. The Fatality Analysis Reporting System (FARS) provides the location of motor vehicle crashes and information

regarding fatalities [17]. Every location of vehicle crashes with fatalities is geocoded. For the period of 2015–2019, the total number of fatality locations is 59,047, and total fatalities in Southeastern region for study years is 63,946.

Each location is treated as a potential demand for emergency medical services, but there are too many numbers of demands for location problem. Too many demand locations of 59,047 can deteriorate the model's performance, and it was necessary to reduce the total number of demand points by applying tessellation grids covering the overall study area. Also, using the same size of tessellation across the study area helps not to ignore the locations without any fatalities up to 2019 as potential demand locations. Because any other locations could potentially have a chance of any severe vehicle crashes occurring even though there were no fatalities at the points across the study years. Creating the tessellation grids was a good way to reduce the number of demand points while covering whole regions of the study area as potential demands.

The basic unit of each tessellation grid in this analysis is the 260-square-mile area of hexagonal grids, with ten miles for each side. The total number of fatalities and other information were assigned to each hexagonal tessellation grid. A total of 2,304 tessellation grids were produced in the study area. Fig 2 presents the spatial distribution of fatality due to motor vehicle

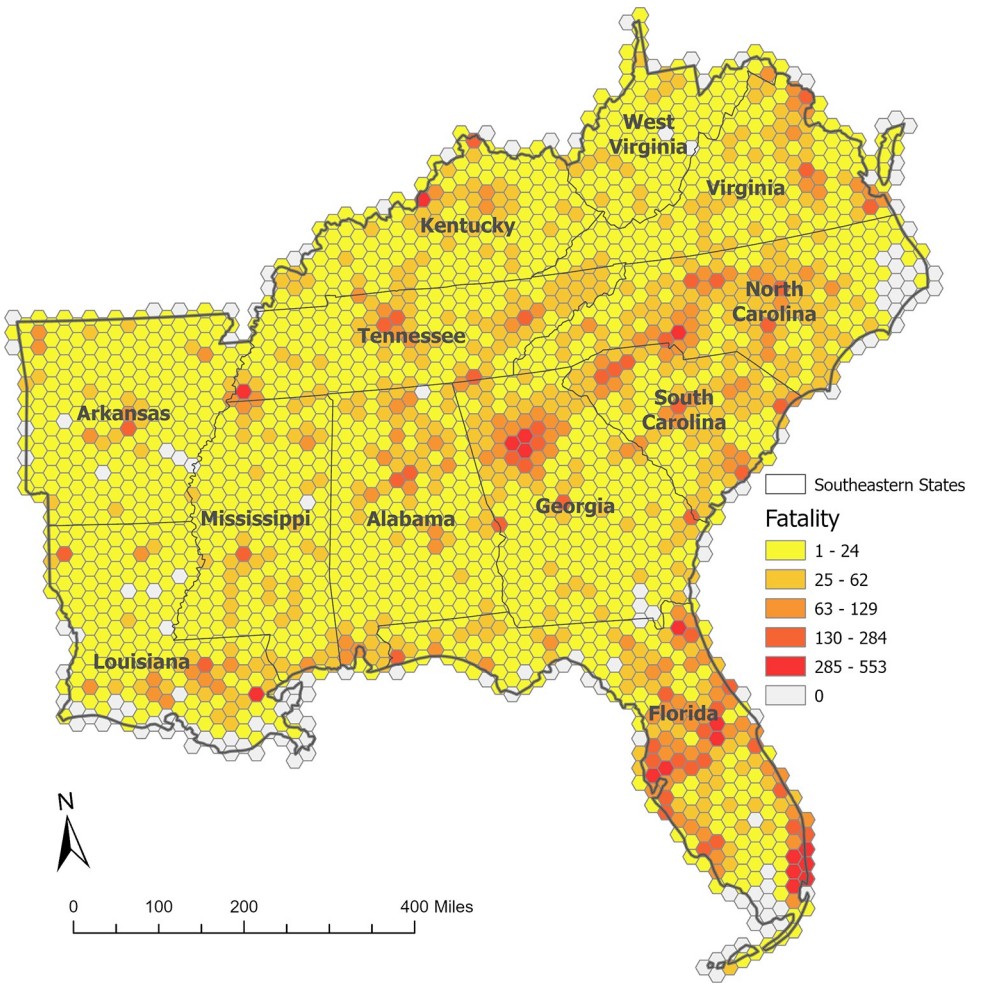

**Fig 2. Potential demands, fatalities, overlaid into the hexagonal tessellations.**

crashes in the study area by hexagonal tessellation grids. The number of fatalities is higher in the grids in and around the urban areas, but there are still many grids with high numbers of fatalities in rural or suburban areas.

## Method

### Finding optimal location of trauma centers: TraCt model

The location problem is used to find the best location for medical facilities [18]. There are many variations in optimal location problems based on the location-allocation problem like *p*-median problem (PMP), and covering models such as maximal covering location problem (MCLP) [19], location set covering problem (LSCP) [20], and anti-covering location problem (ACLP) [21]. Each location problem has different characteristics that help locate diverse types of services [22].

The PMP finds the location of services by assigning demands by pairing one facility to multiple demands, and PMP was utilized for healthcare location problems. However, there are two intrinsic limitations to using PMP to find the optimal locations of EMS or trauma centers. First, PMP assigns the demands to a facility by point-to-point relationship. There are better ways to locate facilities to cover the potential demands inside the aerial service coverage from each candidate. Second, the PMP allocates every demand to one of the selected facilities; however, such an allocation may be infeasible in emergencies due to time sensitivity. The PMP sometimes produces a solution with some demands allocated to the facilities out of the limited travel time distance for the proper treatments for emergency patients [15].

Covering models works with predetermined sets of service locations and their coverage corresponding to the potential demands, and it fits better to locate emergency facilities than the PMP. Two covering models, MCLP and LSCP, represent the covering model, and ACLP is developed to ameliorate the efficiency of traditional covering models. Three covering models pursue different objectives, and here are brief comparisons of those models. The LSCP model tries to cover all demands and tends to locate the number of facilities. Large areas of commonly serving areas would overlap by two or more facilities simultaneously as a solution of LSCP. The MCLP pursues the maximizing demands covered by the restricted number of future facilities. MCLP may leave some demands uncovered if the pre-determined number of facilities is insufficient for complete coverage. The ACLP involves maximizing the number of selected facilities to cover more demands while considering the minimum distance between facilities. This spatial separation constraint directs the model not to choose the facilities within a certain distance. It helps minimize the demands covered by multiple facilities [15]. Thus, the ACLP is covering location problems dealing with equity in terms of EMS by maximizing the whole coverage area by dispersing the facilities and minimizing the demands covered by multiple facilities.

The *TraCt* model is a variant location problem derived from the ACLP to find the optimal location of trauma centers. It maximizes the demands covered by facilities while considering the dispersion constraints of facilities by controlling the minimal distance between them [5]. The *TraCt* model presents the optimal locations for trauma centers in the Southeastern states. The *TraCt* model has three different variations by applying different additional constraints to control the maximum number of facilities accessible from each demand location (*TraCt*-ESC) and the predetermined additional number of total facilities are applicable for the *TraCt* model (*TraCt*-ESCr) while considering existing trauma centers' locations from Chea *et al.* [5]. *TraCt* model is the standard model, and it is applied in this research to compare the result from each

state. The formulation of the TraCt model for this research is presented below:

$$Maximize \sum_k \alpha_k Y_k \tag{1}$$

$$Subject\ to:\ Y_k + Y_j \le 1\ \ \forall k,\ j \in \phi_k, \tag{2}$$

$$Y_k = \{0, 1\}\ \ \forall k. \tag{3}$$

where,

$\alpha_k$: Covered benefit when a candidate or existing trauma facility $Y_k$ is selected

$Y_k$: Index of candidates or current trauma centers

$\Phi_k$: Set of facilities $j$ located inside the facility $k$'s coverage

The objective function (1) is to maximize the sum of potential fatality as demands covered by the TDCs of facilities $Y_k$. Note that $\alpha_k$ represents a weight, a metric of demand, which is covered by the TDC of $Y_k$. The constraints (2) force a facility $k$ to keep separation with any candidate facilities $j$ in the clique set $\Phi_k$. The constraints (3) impose a binary integer restriction on $Y_k$ (1 = if selected, 0 = otherwise).

## Assessing the location of trauma centers: *Phi* correlation coefficient

The *Phi* correlation coefficient is a measure of association for dichotomous variables that occur in a wide range of applications [23–25]. The *Phi* correlation coefficient is useful in many research fields, such as psychology, medicine, and epidemiology. Historically, Pearson used the *Phi* correlation coefficient to study the possible association between vaccination against and recovery from smallpox infection. Pearson also studied the possible associations between the antitoxin serum treatment and recovery from diphtheria [23].

The association of the demand point locations covered by existing facilities and covered by the selection of optimal locations of trauma centers are used to assess current coverage of trauma centers by each demand locations whether covered-or-not situation. The demand locations that covered by any trauma center are presented by 1 and otherwise notated by 0. The association with optimal coverage for demand can be a good measure of assessing the current location of trauma centers. The *Phi* correlation coefficient measures the relationship between covered demand by trauma centers and the *TraCt* model solutions.

*Phi* correlation coefficient (hereafter *Phi* statistic) is a nonparametric statistic used in cross-tabulation with dichotomous values. Whether or not covered by the trauma center can be a variable. Two variables are used depending on whether the trauma center is existing or selected by a solution. For example, a table can be created that consisting of the number of demands covered (1) or uncovered (0) by the current trauma center on the horizontal axis, and the number of demands covered (1) or uncovered (0) by the trauma center selected as a *TraCt* model solution on the vertical axis. *Phi* correlation coefficient measures the strength of an association between two variables. The formulation of *Phi* statistic is presented below.

$$\varphi = \sqrt{\frac{X^2}{n}} \tag{4}$$

where,

$\varphi$: Measure of association

$X^2$: Chi-square of the objective table of covered demands by different conditions.

$n$: Total number of demand locations

The higher *Phi* statistic means a stronger association between the two different demand coverages, and it can be interpreted as demand locations covered by the existing trauma

centers being closer to the demand locations covered by the optimal location of trauma centers chosen by the location problem model.

The lower *Phi* statistic means that there is a lower association between the demand covered by the current facilities and the *TraCt* model solutions. It can be interpreted that the demands covered by existing trauma centers and the solutions are highly different. On the other hand, the higher *Phi* statistic can be interpreted as the less different demand coverage. The vast difference between optimal solutions and existing trauma centers means there is a huge gap to be filled by the improved coverage of future trauma centers for demand locations. In this analysis, the *Phi* statistic can be interpreted as a relative comparison for each state.

## Analysis and results

### Demand locations covered by existing trauma centers by state

The total number and percentage of demand covered by a facility is a simple measure that can be used to represent the level of coverage by trauma centers in the Southeastern region. Comparing the total number of fatalities and the number of demand locations is also an effective way to assess the different demand coverages of each state. Table 1 shows the current coverage of existing trauma centers in each state. Alabama shows the highest percentage of fatalities (5,422/5,604–96.75%) and locations (224/234–95.73%) covered, and Kentucky is the state with the lowest percentage of fatalities (2,133/4,251–50.18%) and locations (70/198–35.35%) covered. The difference in coverage between fatalities (46.57%) and locations (60.38%) is quite substantial, which means the coverage for vehicle crash fatalities is not evenly distributed among states in the Southeastern region. The inherent limitation of this comparison, however, is that the demand locations under the coverage of trauma centers of adjacent states are not considered. The total number of demand locations tends to be underestimated compared to the demand locations that are actually covered by facilities, regardless of whether they are inside or outside the state.

Table 2 shows the current coverage, including the existing trauma centers located in adjacent states, which means sharing state borders with each state. Overall coverage is improved by considering the coverage from the trauma centers of adjacent states. Alabama has the highest

**Table 1. Potential demands covered by the trauma centers in each state of the Southeastern region, 2019.**

| State | Number of TC | Fatalities | | | Number of Demand Locations | | |
|---|---|---|---|---|---|---|---|
| | | Total* | Covered | %** | Total* | Covered | % |
| AL | 51 | 5,604 | 5,422 | 96.75 | 234 | 224 | 95.73 |
| SC | 14 | 5,979 | 5,209 | 87.12 | 148 | 111 | 75 |
| MS | 19 | 4,069 | 3,481 | 85.55 | 225 | 165 | 73.33 |
| GA | 21 | 8,381 | 7,081 | 84.49 | 267 | 174 | 65.17 |
| NC | 17 | 7,843 | 6,423 | 81.89 | 248 | 142 | 57.26 |
| LA | 9 | 3,993 | 3,046 | 76.28 | 233 | 105 | 45.06 |
| FL | 32 | 15,762 | 11,959 | 75.87 | 299 | 171 | 57.19 |
| TN | 13 | 6,007 | 4,485 | 74.66 | 203 | 99 | 48.77 |
| VA | 18 | 4,670 | 3,311 | 70.9 | 212 | 112 | 52.83 |
| WV | 7 | 1,658 | 1,110 | 66.95 | 128 | 62 | 48.44 |
| AR | 19 | 3,380 | 2,173 | 64.29 | 241 | 150 | 62.24 |
| KY | 6 | 4,251 | 2,133 | 50.18 | 198 | 70 | 35.35 |

* The aggregate demand of individual states may be overestimated than the actual total demand of the entire study area.

** The rows of this table are sorted by highest to lowest % of fatalities covered by facilities.

**Table 2. Potential demands covered by the in-state and adjacent trauma centers of the Southeastern region, 2019.**

| State | Number of TC | Fatalities | | | Number of Demand Locations | | |
|---|---|---|---|---|---|---|---|
| | | Total* | Covered | %** | Total* | Covered | % |
| AL | 71 | 5,604 | 5,462 | 97.47 | 234 | 228 | 97.44 |
| SC | 26 | 5,979 | 5,696 | 95.27 | 148 | 127 | 85.81 |
| GA | 43 | 8,381 | 7,576 | 90.39 | 267 | 203 | 76.03 |
| MS | 36 | 4,069 | 3,625 | 89.09 | 225 | 171 | 76 |
| NC | 32 | 7,843 | 6,937 | 88.45 | 248 | 159 | 64.11 |
| AR | 23 | 3,380 | 2,957 | 87.49 | 241 | 171 | 70.95 |
| TN | 29 | 6,007 | 4,955 | 82.49 | 203 | 127 | 62.56 |
| VA | 27 | 4,670 | 3,819 | 81.78 | 212 | 133 | 62.74 |
| LA | 19 | 3,993 | 3,160 | 79.14 | 233 | 119 | 51.07 |
| FL | 48 | 15,762 | 12,199 | 77.4 | 299 | 184 | 61.54 |
| WV | 14 | 1,658 | 1,179 | 71.11 | 128 | 68 | 53.13 |
| KY | 16 | 4,251 | 2,631 | 61.89 | 198 | 91 | 45.96 |

\* The aggregate demand of individual states may be overestimated than the actual total demand of the entire study area.

\*\* The rows of this table are sorted by highest to lowest % of fatalities covered by facilities.

percentage of fatalities (5,462/5,604–97.47%) and locations (228/234–97.44%) covered, and Kentucky has the lowest percentage of fatalities (2,631/4,251–61.89%) and locations (91/198–45.96%). The percentage range decreased for fatalities covered (from 46.57% to 35.58%) and locations covered (from 60.38% to 51.48%) compared to Table 1.

This result shows that no state can cover potential demand locations by using only in-state facilities. Additionally, inequality of accessibility to trauma centers is alleviated in all states when considering the coverage of facilities in-state, and in neighboring states as well because the increased number of demand locations are covered by out-of-state facilities. However, inequality of accessibility to trauma centers among Southeastern states remains, even though the overall coverage improves.

The trauma care dependency on out-of-state facilities of potential demand for each state can be compared using Tables 1 and 2. This should be interpreted as a reference rather than an absolute comparison. Because so many factors can affect the dependency on out-of-state facilities, such as the size and shape of the area of interest, the number of neighboring states, and the total number of accidents and fatalities. Despite this complexity, however, it can be seen that not only in-state but also out-of-state facilities should be considered as candidates for optimal trauma center location modeling. Since timely emergency treatment for trauma patients cannot be fully provided within each state's boundaries, comprehensive cooperation beyond administrative boundaries is essential to utilize facilities that can be reached within a golden hour from areas where accidents may occur.

## Demand locations covered by optimal locations of trauma centers by state

Given candidate facilities for each state in the Southeastern region, the basic *TraCt* model can be used to find the optimal location for trauma centers. Fig 3 displays the solutions the *TraCt* model identified for each state and reveals that the optimal locations are more evenly distributed than the existing trauma centers—though some demand points are not covered by optimal locations for trauma centers. Those points are demand locations that do not include any fatality or motor vehicle crashes on roads, such as national parks, state parks, large lakes, or wetlands—thus, the model does not recognize them as areas to be covered.

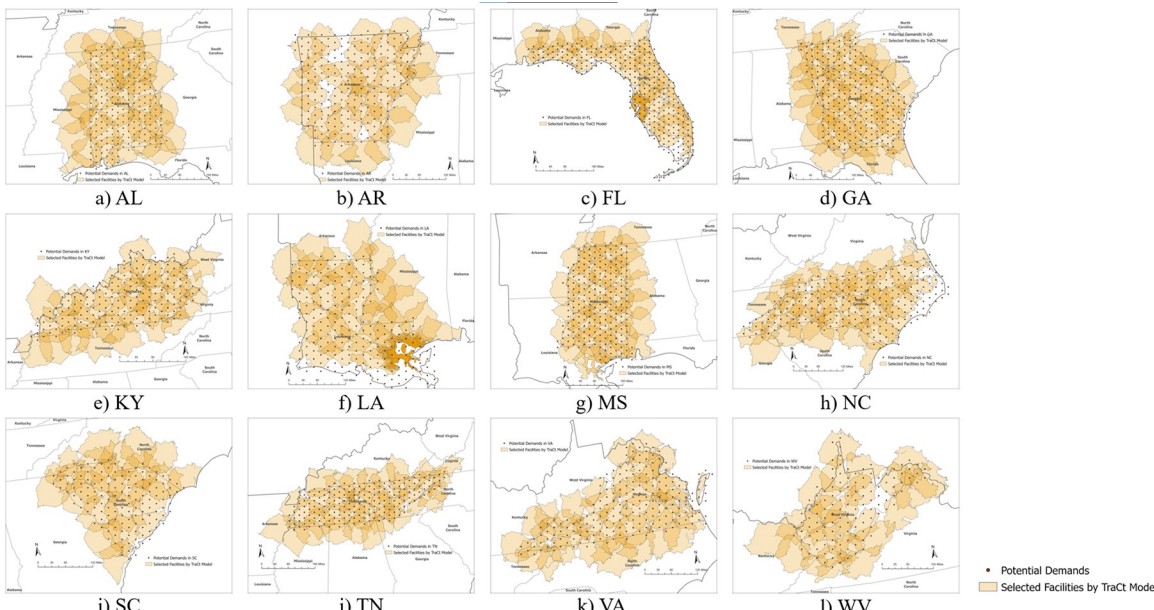

**Fig 3. Optimal location of trauma centers selected by the *TraCt* model for each Southeastern state.**

Table 3 lists the potential demands covered by the *TraCt* model solution in the Southeastern regions. The total percentage of potential demand covered by optimal locations of facilities increased to higher than 80.1% and 93.7% in most of the Southeastern states. Kentucky was the state with the highest percentage of fatalities (4,236/4,251–99.65%) and locations (191/198–96.46%) covered, and Florida showed the lowest percentage of fatalities (12,625/15,762–80.1%) and demand locations (222/299–74.25%) covered. The range of percentage of covered demands decreased for both fatalities (from 35.58% to 19.55%) and demand locations covered (from 51.48% to 22.21%), compared to the demand coverage of existing trauma centers in 2019.

**Table 3. Potential demands covered by the *TraCt* model solutions in the Southeastern region.**

| State | Number of TC | Fatality | | | Number of Demand Locations | | |
|---|---|---|---|---|---|---|---|
| | | Total* | Covered | %** | Total* | Covered | % |
| KY | 31 | 4,251 | 4,236 | 99.65 | 198 | 191 | 96.46 |
| TN | 37 | 6,007 | 5,969 | 99.37 | 203 | 198 | 97.54 |
| GA | 44 | 8,381 | 8,327 | 99.36 | 267 | 261 | 97.75 |
| MS | 32 | 4,069 | 4,011 | 98.57 | 225 | 218 | 96.89 |
| SC | 23 | 5,979 | 5,859 | 97.99 | 148 | 138 | 93.24 |
| NC | 34 | 7,843 | 7,654 | 97.59 | 248 | 212 | 85.48 |
| AR | 28 | 3,380 | 3,296 | 97.51 | 241 | 223 | 92.53 |
| WV | 25 | 1,658 | 1,614 | 97.35 | 128 | 120 | 93.75 |
| AL | 36 | 5,604 | 5,422 | 96.75 | 234 | 227 | 97.01 |
| LA | 34 | 3,993 | 3,828 | 95.87 | 233 | 182 | 78.11 |
| VA | 37 | 4,670 | 4,380 | 93.79 | 212 | 187 | 88.21 |
| FL | 38 | 15,762 | 12,625 | 80.1 | 299 | 222 | 74.25 |

\* The aggregate demand of individual states may be overestimated than the actual total demand of the entire study area.

\*\* The rows of this table are sorted by highest to lowest % of fatalities covered by facilities.

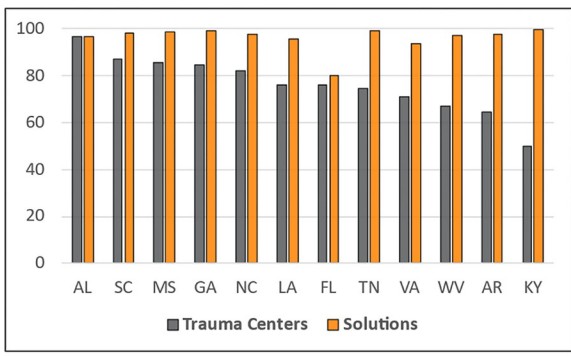
a) Covered fatalities by facilities

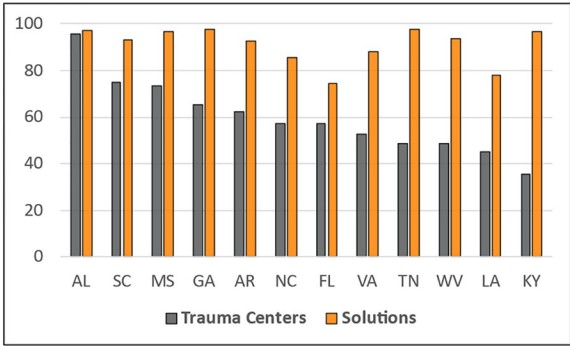
b) Covered locations by facilities

**Fig 4. Percentage of covered demands by state.**

Fig 4 compares the differences in the percentage of demand covered by existing trauma centers and model solutions. The *TraCt* model solutions cover more potential demands compared to existing trauma centers. Some states, such as Kentucky, Arkansas, West Virginia, Virginia, and Tennessee, show incredible improvements in demand coverage in fatalities and demand locations.

The *Phi* statistic helps to compare the demand coverage of current locations with the optimal locations produced by the *TraCt* model in a quantitative comparison. The *Phi* statistic is presented as an index ranging from 0 to 1, without negative values. A *Phi* statistic closer to 1 indicates there is a stronger association between the two variables. The statistical significance can be measured by *p*-value of chi-square test.

Fig 5 shows the *Phi* statistics for each state. A *p*-value less than 0.05 is typically considered to be statistically significant, and it means there is a correlation between the two variables in the chi-square test. In this study, four Southeastern states—Kentucky, Arkansas, Tennessee, and West Virginia—showed a *p*-value less than 0.05, and there was no significant correlation between the coverage of trauma centers in 2019 and the coverage of trauma centers chosen as a solution to the *TraCt* model. The lack of statistical significance in the chi-square test can be interpreted in a negative way because the results of the *TraCt* model produce optimal coverage of the chosen trauma center. The absence of statistical significance or the existence of a lower level of correlation with an optimal situation is less favorable than a higher correlation.

There are no strict standards to interpret the value of *Phi* statistic, the level of association, but, in many empirical studies, a value above 0.25 is considered a strong association [24]. In this study, instead of trying to interpret the absolute value, this result was used to compare the degree of difference among the 12 Southeastern states. Because the lower *Phi* statistic indicates there is a gap between the current location and optimal solutions, the states with lower *Phi* statistic are in greater need of improvement in terms of location and spatial distribution of the trauma centers. Florida has the highest *Phi* statistic (0.608), and Louisiana (0.494), Alabama (0.487), North Carolina (0.439), South Carolina (0.404), Virginia (0.396), and Mississippi (0.301) follow, with a relatively strong association between the coverages of the current trauma center and optimal locations of the trauma center.

Table 4 shows the number of facilities chosen by the *TraCt* model in-state or out-of-state. It shows the trauma care dependency toward out-of-state candidates owing to the number of selected facilities in contiguous states. For example, only 20 (20/37–54.1%) trauma centers are located inside Tennessee's administrative boundary, whereas 17 (17/37–45.9%) of the trauma centers are located outside but adjacent to Tennessee. The percentage of trauma centers

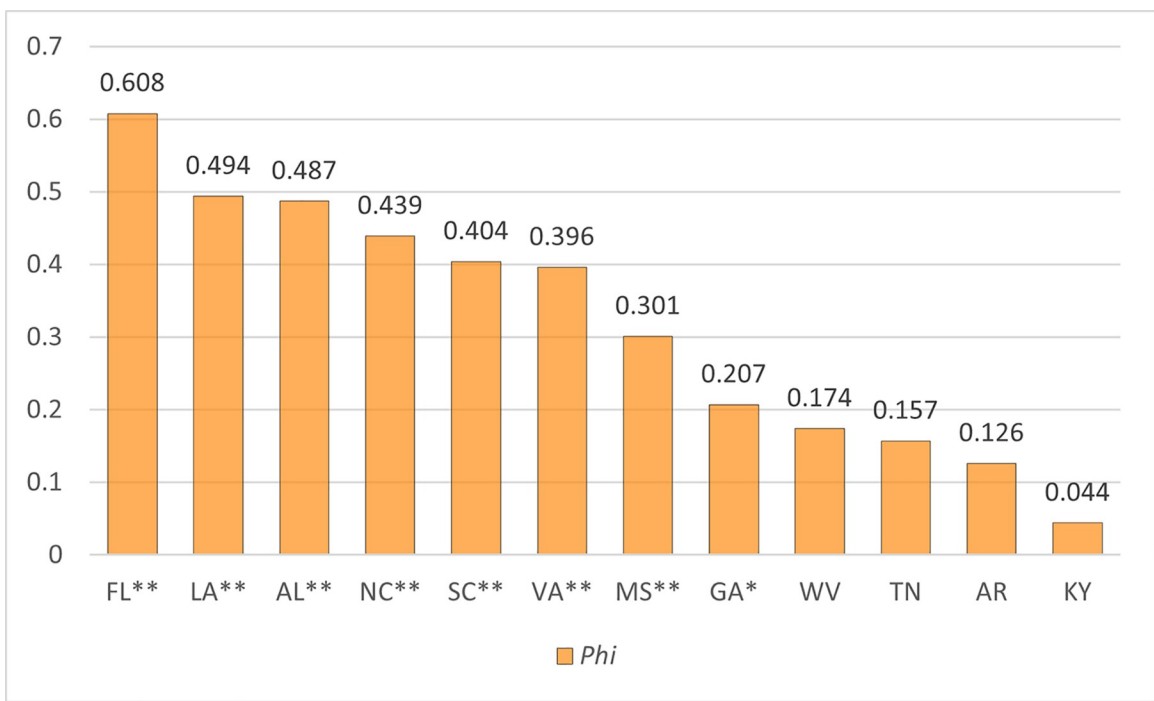

**Fig 5. *Phi* correlation coefficient between coverage of trauma centers by current locations and optimal solutions from the *TraCt* model.**

located out of states well corresponds with the number of contiguous states that share state borders. For example, Florida has the lowest percentage of out-of-state facilities due to most borderline being on the seashore without neighboring states. And the percentage of out-of-state facilities and the number of contiguous states have a strong positive correlation in the Pearson correlation coefficient, $r(10) = 0.606$, $p = 0.037$. The result is significant at $p < 0.05$.

**Table 4. Number of facilities by location, whether within or outside the state borders.**

| State | Total | In State | | Out State | | Number of outside states by each state |
|---|---|---|---|---|---|---|
| | | Count | %* | Count | % | |
| FL | 38 | 30 | 78.9 | 8 | 21.1 | 2 (2 –AL, GA) |
| LA | 34 | 25 | 73.5 | 9 | 26.5 | 2 (7 –MS; 2 –AR) |
| AR | 28 | 20 | 71.4 | 8 | 28.6 | 3 (4 –LA; 3 –MS; 1 –TN) |
| NC | 34 | 22 | 64.7 | 12 | 35.3 | 4 (5 –SC; 3 –TN, VA; 1 –GA) |
| WV | 25 | 16 | 64 | 9 | 36 | 2 (7 –VA; 2 –KY) |
| GA | 44 | 27 | 61.4 | 17 | 38.6 | 5 (6 –AL; 4 –FL; 3 –SC; 2 –NC, TN) |
| MS | 32 | 19 | 59.4 | 13 | 40.6 | 4 (5 –AL; 3 –LA, TN; 2 –AR) |
| KY | 31 | 18 | 58.1 | 13 | 41.9 | 4 (8 –TN; 2 –VA, WV; 1 –AL) |
| VA | 37 | 21 | 56.8 | 16 | 43.2 | 4 (6 –WV; 5 –NC; 3 –TN; 2 –KY) |
| SC | 23 | 13 | 56.5 | 10 | 43.5 | 2 (7 –NC; 3 –GA) |
| TN | 37 | 20 | 54.1 | 17 | 45.9 | 6 (4 –KY; 3 –AL, AR, NC; 2 –MS, VA) |
| AL | 36 | 19 | 52.8 | 17 | 47.2 | 4 (7 –MS; 5 –TN; 3 –GA; 2 –FL) |

\* The rows of this table are sorted by highest to lowest % of in-state facilities.

Finding optimal locations of trauma centers for each state results in great improvements in terms of geographic coverage. However, putting together a list of facilities in each state across the Southeastern region reveals that many more facilities have been selected than are actually needed in the Southeastern region. Many facilities near state borders are selected to cover only a few demands. This is because the location of out-of-state trauma centers does not consider the actual demand from their own state when the *TraCt* model was applied at the individual state level. In short, even though there are great improvements in potential demand coverage in solving the location problem for the individual state level, there is an issue where too many trauma centers are selected for the entire Southeastern region. As *TraCt* model found the optimal location of trauma centers in all individual States, it implies that the *TraCt* model will be valid to find the optimal location of trauma centers in other cases. The *TraCt* model will be a helpful tool for finding the optimal Trauma center locations in any area without a trauma care system or areas with abysmal facility distribution.

## Analysis of the entire Southeastern region

The *TraCt* model was applied to potential demand locations of the entire Southeastern region to find the minimum number of trauma centers needed to cover the maximum potential demand in the overall study area. The solution from the *TraCt* model for the overall study area of the Southeastern region shows much better coverage for potential demand compared to existing total coverage of trauma centers—but slightly lower coverage compared to the sum of *TraCt* model solutions for individual states. As a result of the *TraCt* model with the fatality, a total of 241 chosen facilities covered around 98.37% (62,904 fatalities) of potential demand. Compared to the current coverage of 88.9% (56,846 fatalities) of potential demand from the 226 current trauma centers, the 15 additional facilities helped improve the overall coverage by 6,058 fatalities (9.47%) when the *TraCt* model was applied to the entire Southeastern region. Applying the model to individual states showed a slightly higher percentage (98.69%) of covered demands (63,109 fatalities) with the 344 trauma centers selected. Although the aggregation of individual state models covers 205 fatalities more than the entire Southeastern model, it is not efficient, considering that a numerous 103 additional facilities are required for this. The summary of the different models is shown in Table 5.

The entire spatial extent model of the Southeastern region is effective for minimizing excessive trauma center designation problems. Fig 6 presents the spatial distribution of current trauma centers and the results of the *TraCt* model solutions, along with the number of facilities located in each state. The model solution provides the greatest coverage, with a more evenly distributed number of facilities. Enlarging the scale of the study area's spatial extent works well by reducing the number of contiguous states and simplifying the study area's boundary shape in the Southeastern region. This spatial extent scale effect helps the *TraCt* model applied for the larger area find the optimal location of facilities more efficiently than the smaller spatial extents applied model.

**Table 5. Summary for three different model's number of facilities and demand coverages.**

| Facilities | Number of Facilities | | Fatalities | | | Locations | | |
|---|---|---|---|---|---|---|---|---|
| | Total | Duplicated | Total | Covered | % | Total | Covered | % |
| Trauma Centers 2019 | 226 | - | 63946 | 56846 | 88.90 | 2304 | 1620 | 70.31 |
| TraCt–Individual State | 344 | 50 | | 63109 | 98.69 | | 2171 | 94.23 |
| TraCt–Southeastern Region | 241 | - | | 62904 | 98.37 | | 2163 | 93.88 |

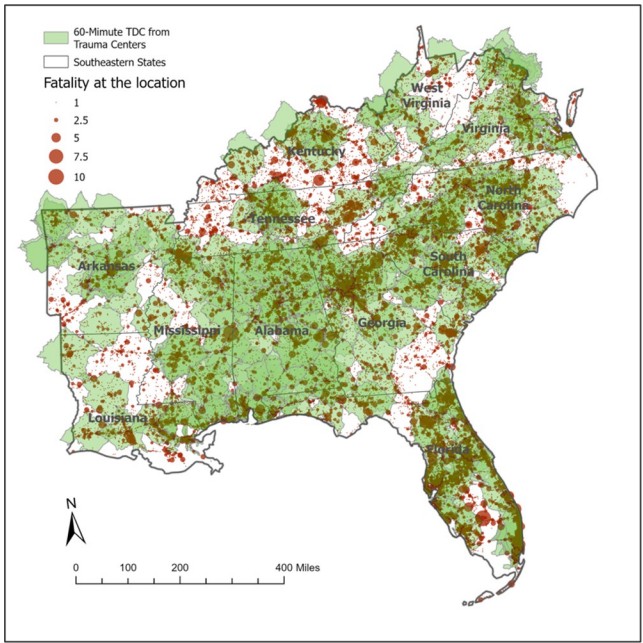
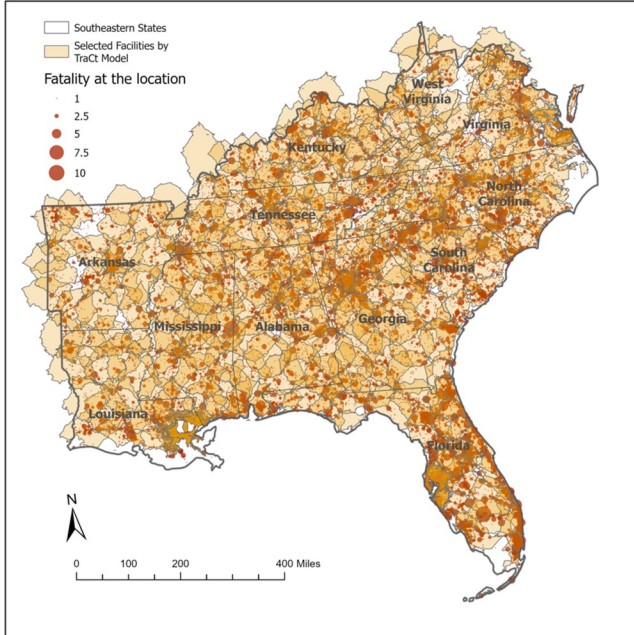

a) Spatial distribution of current trauma centers

b) Spatial distribution of trauma centers by *TraCt*

**Fig 6. Spatial distribution of trauma centers in 2019 and selected by the *TraCt* model for the Southeastern region.**

The number of trauma centers in 2019 is quite uneven across the Southeastern states. While Kentucky has only six trauma centers, Alabama has 8.5 times more trauma centers than Kentucky, with 51 facilities. Fig 7 shows the different number of trauma centers for each state by different model applications: current trauma centers of 2019, the aggregation of the result of *TraCt* models for individual states, and the results of the *TraCt* model for the entire Southeastern region. The number of trauma centers in 2019 is quite uneven across the Southeastern states. It is not fair because both states have not that different in the number of fatalities, 5,604 in Alabama and 4,251 in Kentucky. When the *TraCt* model is applied to each state, the considerable gap between the state with the highest and lowest number of trauma centers is filled with fewer trauma centers in the states with excessive supplies and more facilities for the states with fewer trauma centers in 2019.

The optimal solution from the *TraCt* model for the Southeastern region is a spatially dispersed facility distribution pattern compared to the location of existing trauma centers, and it has the strength to designate a relatively fair number of trauma centers in each state compared to other models. The heights of the bar graph of the *TraCt* model for the entire region are most evenly distributed compared to any other colored bar in Fig 7. Enlarging the spatial scale of the study area of the *TraCt* model is a way to avoid excessive designation of trauma centers in the overall region.

## Conclusion

This study was aimed at measuring the spatial disparities of trauma center accessibility in the Southeastern region of the United States, and it suggests policy implications based on the research outcomes of the *TraCt* model and comprehensive regional area approach. The empirical research showed three notable analysis results: First, by comparing the level of covered demands in the 12 states of the Southeastern region, this study showed that much potential

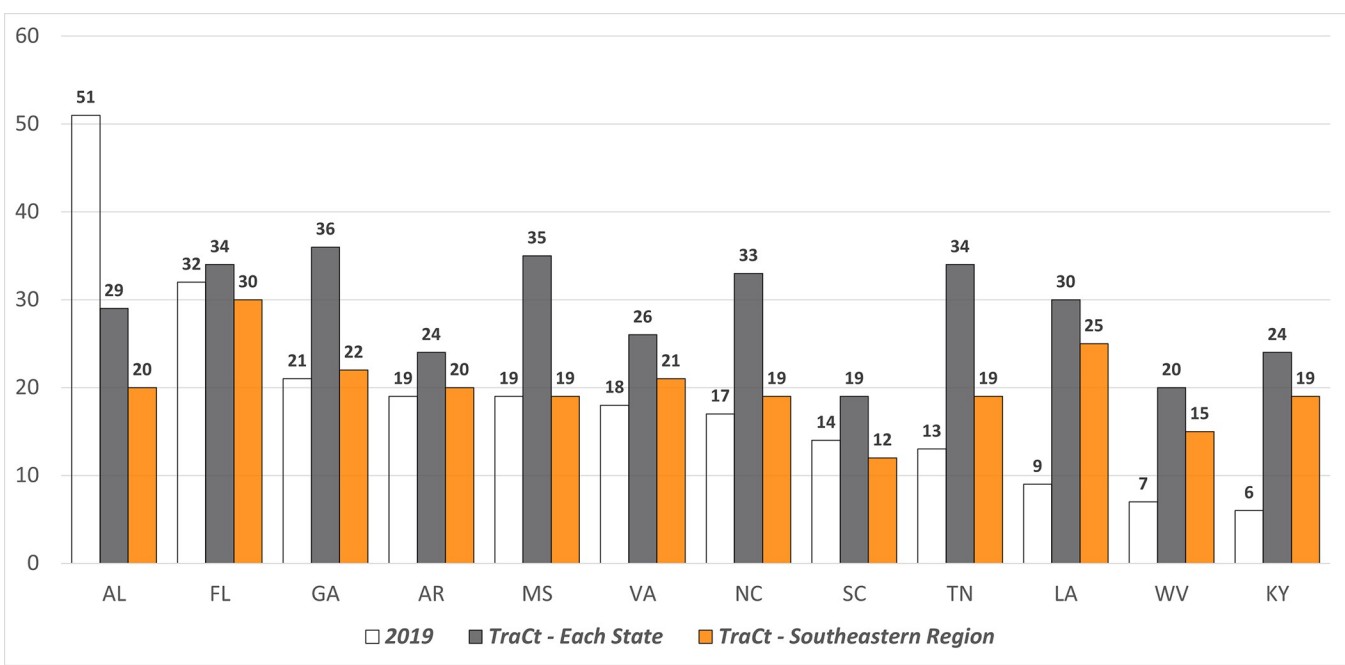

**Fig 7. Number of trauma centers by different model application.**

demand remained, where there was no chance to receive opportune trauma medical services. Second, the *TraCt* model found the optimal locations of trauma centers, with higher level of demands coverage. The optimal location of trauma centers, including the candidates outside of state boundaries, was also found using the *TraCt* model. This approach revealed that out-of-state facilities helped improve demand coverage for each state, including the spatial dependency of out-of-state trauma centers. Applying the *TraCt* model for each state increased the overall coverage. However, this approach presented too large number of facilities for each state, without considering the total demand located adjacent to the state in terms of the wider spatial extent. Third, the model was applied to the entire Southeastern region. Ultimately, the broader regional application of the *TraCt* model increases total coverage while lessening the total number of facilities in the entire study area. Additionally, the *TraCt* model for the Southeastern region reduced spatial inequality in regard to accessing trauma centers by providing a dispersed spatial distribution of optimal locations of trauma centers and a fair number of facilities in all individual states.

Three policy implications are derived from the result of empirical analysis. First, cooperation exceeding the administrative boundary is necessary to transfer the maximal number of trauma patients to trauma centers in time for definitive care. None of the municipal governments can cover all potential trauma care demands alone, and considering the actual location of potential demands can help coordinate with contiguous municipal governments. Second, the *TraCt* model works well with maximum coverage of potential demands. Even though this analysis is for the Southeastern region of the U.S., the multifarious *TraCt* model can be applied to any other region or nation or based on the different kinds of potential demands. Third, enlarging the scale of the study area's spatial extent beyond the single administrative boundaries helps the *TraCt* model find the most efficient number of facilities while covering the maximal number of potential demands. The *TraCt* model for the larger scale of spatial extent works better for some countries without any trauma center system settled. Additionally, the

*TraCt* model is more powerful when applied to regions or nations without comprehensive trauma facility management systems or regions with not too many trauma centers located primarily for the first phase of the plan to locate trauma centers.

However, this research has two limitations on the *TraCt* model to be ready for practical application in the actual situation at this point. First, this study only considered the fatalities caused by vehicle accidents, but many fatalities are still due to unexpected diseases and accidents. The *TraCt* would be much more credible when it takes diverse potential demands into account the model. Second, the *TraCt* model is static because this version of *TraCt* utilizes a single coverage area for each candidate. The service coverage area is the most critical factor that works with the model, and the ground truth is that it varies according to the hours, days, and even seasons due to the dynamic flow of traffic on the road network. The spatiotemporal approach would be suitable for creating dynamic outcomes corresponding to the real world. Future research aims to improve the *TraCt* model to be more comprehensive and dynamic enough to be applied to the real world. For future research, thoroughly investigating the diverse datasets of potential demands helps the TraCt model be pragmatic. The spatiotemporal dynamic model research and application development will enable emergency medical services that can respond to changes in time and space.

The practical effort should be preceded to facilitate future research and develop practical applications of the *TraCt* model in a short time. Sharing data and applying innovative methods to find the optimal spatial extent across multiple counties or states are most important to improve the quality of EMS services. Promoting coalitions encompass diverse agents, such as healthcare and trauma-care professionals, policymakers, planners, geographers, researchers, and even engineers, can be a cornerstone for better emergency medical services. It will be very helpful in developing a more advanced *TraCt* model for faster accessibility to the trauma centers if different agents can cooperate.

## Author Contributions

**Conceptualization:** Heewon Chea, Hyun Kim.

**Data curation:** Heewon Chea.

**Formal analysis:** Heewon Chea.

**Investigation:** Heewon Chea.

**Methodology:** Heewon Chea, Hyun Kim.

**Project administration:** Hyun Kim.

**Resources:** Heewon Chea.

**Software:** Heewon Chea.

**Supervision:** Hyun Kim.

**Validation:** Hyun Kim.

**Visualization:** Heewon Chea.

**Writing – original draft:** Heewon Chea.

**Writing – review & editing:** Heewon Chea, Hyun Kim.

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
