## [Decision Letter · Decision Letter 0]

14 Mar 2023

PGPH-D-22-01274

Assessing trauma center accessibility in the Southeastern region of the U.S. to improve healthcare efficacy using an Anti-covering approach

Dear Dr. Chea,

Thank you for submitting your manuscript to PLOS Global Public Health. After careful consideration, we feel that it has merit but does not fully meet PLOS Global Public Health’s publication criteria as it currently stands. Therefore, we invite you to submit a revised version of the manuscript that addresses the points raised during the review process.

Overall, the reviewers were impressed with your work but ask for more explanation and clarification of the methodology used to be more understandable to the average global surgery/global health researcher.  They also indicate that you would do well to expand upon the implications of this work in the policy space.  

While the text is readable, prior to acceptance we require a thorough and careful proofread for grammar and sentence structure.  Throughout the manuscript there are multiple grammatical errors that require revision.  Please pay close attention to this.  

We look forward to receiving your revised manuscript.

Kind regards,

Rashi Jhunjhunwala

Academic Editor

Journal Requirements:

Additional Editor Comments (if provided):

Dear Authors,

Thank you for submitting this important work to our Journal. Overall, this is an impressive and important contribution to the literature and utilizes novel and accessible methodologies to address a clinically pertinent issue in trauma care. Further, we appreciate the selection of your study area as it is a classically underserved geography with few trauma centers and reasonably high volume of injury.

A few minor comments:

1) Please review the manuscript very closely to address grammatical errors. Some of these from the first section are detailed below:

1- Line 80: Suggest revision to: Prior research has been focused...

2- Lines 81-82: Suggest revision to: Some of this research has focused on the Southeastern region of the U.S., and found that it has the worst trauma accessibility country-wide.

3- Lines 82-83: suggest revision to: "However, despite these insights, this research overlooks the fact that even...

4- Line 90: This sentence should be located in the same paragraph as the previous sentence.

5- Line 96: Suggest revision to: However, little research has been done...

6- Lines 101-102:Confusing phrasing. Needs revision

There are multiple instances where the meaning is lost due to unclear syntax and/or grammar. These must be resolved if the manuscript is to be accepted for publication.

2) It would be helpful to the reader if your tables were arranged such that there was a clear order to the way in which the states are displayed-- for example, if they were ranked by highest to lowest number of trauma centers, % fatalities covered, % demand locations covered, etc. Currently, there is no clear logic to the order in which the states' data is displayed which makes it confusing to understand rank order or to compare between states.

3) The image quality of your graphics should be higher resolution so that the legends are more easily read.

Reviewers' comments:

Reviewer's Responses to Questions

**Comments to the Author**

1. Does this manuscript meet PLOS Global Public Health’s publication criteria? Is the manuscript technically sound, and do the data support the conclusions? The manuscript must describe methodologically and ethically rigorous research with conclusions that are appropriately drawn based on the data presented.

Reviewer #1: Yes

Reviewer #2: Yes

2. Has the statistical analysis been performed appropriately and rigorously?

Reviewer #1: Yes

Reviewer #2: Yes

3. Have the authors made all data underlying the findings in their manuscript fully available (please refer to the Data Availability Statement at the start of the manuscript PDF file)?

Reviewer #1: Yes

Reviewer #2: Yes

4. Is the manuscript presented in an intelligible fashion and written in standard English?

Reviewer #1: Yes

Reviewer #2: Yes

5. Review Comments to the Author

Reviewer #1: This is an interesting paper that uses the Anti-covering location model for trauma centers (TraCt model) which involves maximizing the number of facilities without placing 2 within a certain seperation standard. The authors have published this for Tennessee (Chea H, Kim H, Shaw SL, Chun Y. Assessing Trauma Center Accessibility for Healthcare Equity Using an Anti-Covering Approach. Int J Environ Res Public Health. 2022;19(3):1459. Published 2022 Jan 27. doi:10.3390/ijerph19031459) and are extending this to the South East. IT will practically help in the efficient location of trauma centers across state lines if this model is embraced.

The conclusions are robust and clear, and address the objectives of the paper. Recommendations are practical and efficient. The Global Surgical care premises are appropriate.

However, from the early abstract to the methods, the TraCt model methods are not clarified to the uninitiated, lay global health reader without a background in statistics, or in previous literature. ACLP for a trauma center location problem should be clearly defined and described in the methods. Clarity can be added through the text and more needs to be done in clearly defining the modeling that was used earlier in the manuscript, and explaining some background to the modeling to ground this in content clarified this paper, beyond a reference.

Reviewer #2: The article presents a well-structured and coherent narrative of the empirical results and policy implications of the study. However, it could benefit from a more nuanced approach that emphasizes the key contributions and future directions of the research.

To this end, it is crucial to highlight the innovative approach of the Anti-covering location problem (ACLP) used to assess trauma center accessibility, which represents a significant step forward in the field of healthcare efficacy. The use of the TraCt model to identify optimal locations of trauma centers is another vital contribution that cannot be overlooked. Moreover, the application of a comprehensive regional area approach that addresses spatial disparities in accessing trauma centers further solidifies the research's impact.

Moving on to the most significant findings of the study, it is crucial to underline the potential demand that remains unaddressed in the Southeastern region of the U.S. The identification of optimal locations of trauma centers through the TraCt model and the subsequent reduction of spatial inequality in accessing trauma centers represent a critical achievement that cannot be overstated.

However, it is important to acknowledge the limitations of the study and explore future directions for research that can enhance its impact. For instance, incorporating more data sources and analytical techniques, considering other factors such as socio-economic status and transportation infrastructure, and expanding the spatial extent of the study area are all viable options that could significantly improve the research's overall value.

Lastly, practical recommendations for policymakers and healthcare professionals must be presented based on the study's findings. Highlighting the need for cooperation between county governments to transfer trauma patients to trauma centers in a timely fashion, utilizing the TraCt model to identify optimal locations of trauma centers, and emphasizing the importance of considering the spatial extent of the study area in addressing spatial disparities in accessing trauma centers are all crucial steps that can be taken to improve healthcare efficacy in the region.

6. PLOS authors have the option to publish the peer review history of their article (what does this mean?). If published, this will include your full peer review and any attached files.

**Do you want your identity to be public for this peer review?** For information about this choice, including consent withdrawal, please see our Privacy Policy.

Reviewer #1: No

Reviewer #2: **Yes: **Arturo Cervantes Trejo

---

## [Editor Report · Decision Letter 1]

26 May 2023

PGPH-D-22-01274R1

Assessing trauma center accessibility in the Southeastern region of the U.S. to improve healthcare efficacy using an Anti-covering approach

Dear Dr. Chea,

Thank you very much for submitting your revisions and addressing the comments made by our reviewers. While this manuscript is very close, it still requires more copyediting. I would recommend reviewing this resource: https://plos.org/resource/how-to-edit-your-work/ and consider utilising this resource https://plos.editage.com/. There is a discount for PLOS authors.

We look forward to your next revision.

We look forward to receiving your revised manuscript.

Kind regards,

Rashi Jhunjhunwala

Academic Editor
---

## [Editor Report · Decision Letter 2]

10 Jul 2023

Assessing trauma center accessibility in the Southeastern region of the U.S. to improve healthcare efficacy using an Anti-covering approach

PGPH-D-22-01274R2

Dear Dr. Chea,

We are pleased to inform you that your manuscript 'Assessing trauma center accessibility in the Southeastern region of the U.S. to improve healthcare efficacy using an Anti-covering approach' has been provisionally accepted for publication in PLOS Global Public Health.

Best regards,

Rashi Jhunjhunwala

Academic Editor